# Eggs for Improving Nutrition, cognitive development and reducing linear growth retardation among Infants and young Children (ENRICH): protocol of an egg supplementation trial among children aged 9–18 months in Hyderabad, India

Santosh Kumar Banjara [1], Sai Ram Challa [1], Little Flower Augustine,[1] Teena Dasi,[1] Radhika Madhari,[1] Sylvia Fernandez Rao,[1] Ravindranadh Palika,[1] Raghu Pullakhandam [1], Rajender Rao Kalashikam,[1] Ramachandrappa Naveen Kumar [1], Dharani Pratyusha Palepu [1], Monica Chilumula,[1] Kiruthika Selvaraj,[1] Preethi Gopinath [1], Hilary Davies-Kershaw [2], Modou Lamin Jobarteh [3], Paul Haggarty,[4] Claire Heffernan,[5,6] Elaine Ferguson,[7] Bharati Kulkarni[1]

For numbered affiliations see end of article.

**Correspondence to**
Dr Sai Ram Challa; drchallanin@gmail.com

## ABSTRACT

**Introduction** Evidence on the impact of nutrient-rich animal source foods such as eggs for improving child growth and cognition is inconsistent. This study aims to examine the impact of an egg intervention in children, along with behaviour change communication (BCC) to the mother, on linear growth and cognition, and nutritional status in children aged 9–18 months.

**Methods and analysis** A 9-month open-labelled randomised controlled trial will be conducted in three urban slums in Hyderabad, India, as a substudy of an observational cohort study (n=350) following pregnant women and their children until 18 months of age in a population at risk of stunting. The children born to women enrolled during the third trimester of pregnancy will be block randomised in a 1:4 ratio into the intervention (n=70) and control (n=280) groups. Children in the intervention group will be supplemented with one egg per day starting from 9 months until 18 months of age. BCC designed to enhance adherence to the intervention will be used. The control group will be a part of the observational cohort and will not receive any intervention from the study team. The primary outcome will be length-for-age z-scores, and the secondary outcomes will include cognition, blood biomarkers of nutritional status including fatty acid profile and epigenetic signatures linked with linear growth and cognition. Multivariate intention-to-treat analyses will be conducted to assess the effect of the intervention.

**Ethics and dissemination** The study is approved by the Institutional ethics committees of ICMR-National Institute of Nutrition, Hyderabad, India and London School of Hygiene and Tropical Medicine, UK. The results will be published in peer-reviewed journals and disseminated

### WHAT IS ALREADY KNOWN ON THIS TOPIC

⇒ Previous studies using single or multiple micronutrients or cereal legume-based food supplements have shown only modest impact on child stunting.
⇒ Evidence on the impact of nutrient dense animal source foods such as eggs for improving child growth and cognition is inconsistent.

### WHAT THIS STUDY ADDS

⇒ Comprehensive evidence on the impact of daily egg supplementation during 9–18 months of age on the linear growth, cognition, nutritional biomarkers and epigenetic outcomes.

### HOW THIS STUDY MIGHT AFFECT RESEARCH, PRACTICE OR POLICY

⇒ Eggs provide several critical nutrients to support the infant growth and development and egg supplementation is a promising option for nutrition programmes.
⇒ Therefore, the trial has a potential for immediate scale-up.

to policy-makers. Findings will also be shared with study participants and community leaders.
**Trial registration number** CTRI/2021/11/038208

## INTRODUCTION

The high prevalence of stunting in children under 5 years of age in India (35%) remains a cause of concern given the associated

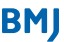

long-term negative consequences on cognition, physical work capacity, earning potential and risk of chronic diseases in adulthood.[1–4] Childhood stunting is common among populations residing in adverse environments with limited access to nutrient-dense foods and poor sanitation.[5] In these environments, children's diets are predominantly cereal-based, with low amounts of animal source foods (ASFs) that contain high-quality complete protein, and do not meet the high nutrient needs of young children.[6] Improving the quality of early childhood diets and reducing child stunting is therefore a public health priority in India.

Previous evidence based on randomised controlled trials (RCTs) evaluating supplementation with cereal legume-based complementary foods or single or multiple micronutrient interventions in children has shown a limited impact on linear growth.[7–9] The emerging evidence suggests that locally available ASFs such as eggs could play an important role in improving linear growth of young children.[10] However, the evidence is mixed, and the impact of an egg intervention on linear growth and stunting prevalence appears to vary in different settings depending on the dietary patterns and baseline prevalence of child stunting. For example, an RCT in 163 children (6–9 months) from Ecuador assessed the impact of egg supplementation (one egg per day for 6 months) and showed a reduction in the prevalence of stunting by 47% and underweight by 74% in the intervention arm (n=83).[11] However, in a similar trial in 660 children (6–9 months) from Malawi, an egg intervention did not have a beneficial effect on linear growth, possibly due to low baseline prevalence of stunting and background diets being rich in ASFs.[12] In another RCT from Malawi (9-month-old children, N=267), the addition of egg powder with bovine colostrum to complementary feeds resulted in lower prevalence of stunting in the intervention arm (n=135, RR=0.70; 95% CI=0.52 to 0.94).[13] Interventions such as preventive small-quantity lipid-based nutrient supplements for children aged 6–23 months, have shown positive impact on child growth.[14] Compared with such interventions, a locally available ASF such as egg will be cost-effective and scalable.[15] The aetiology of stunting is complex and poorly understood and the effect of interventions may differ between countries and ethnicities. There is no evidence available from the Indian subcontinent, where the stunting prevalence is high, and the ASF intake is low.

Eggs provide several nutrients critical to infant growth and development which include all the essential amino acids making it a high-quality/complete protein, docosahexaenoic acid (DHA) and micronutrients such as vitamin $B_{12}$ which are important for neural development and immune function.[10] Eggs are among the richest sources of choline, which is necessary for the production of phospholipids, cell membrane integrity and the conversion of acetylcholine and sphingomyelin necessary for brain development and function.[16] Dietary choline intake can modulate DNA methylation, therefore, egg

supplementation may also impact the epigenetic signatures.[17] Epigenetics is one of the candidate mechanisms linking early life factors to physical and cognitive development with consequences across the life course.[18–21] Evaluation of epigenetics signatures linked with stunting is central to the Action Against Stunting Hub (AASH) observational cohort.[22]

In India, eggs are more affordable than other ASFs and are relatively simple to store and prepare. Egg supplementation is a promising option in government nutrition programmes. In recent years, a few states in India have included eggs in supplementary feeding programmes for pregnant women and children in the age group 7 months to 6 years.[23 24] However, systematic impact evaluation of these interventions on anthropometric, biochemical or cognitive outcomes has not been done. We hypothesise that daily egg supplementation (accompanied by behaviour change communication (BCC) to increase compliance to the intervention) starting at 9 months will improve linear growth and reduce the incidence of stunting at 18 months of age.

The proposed Eggs for Improving Nutrition, cognitive development and reducing linear growth retardation among Infants and young Children (ENRICH) study aims to evaluate the effect of daily supplementation with an egg in children from 9 months to 18 months of age (accompanied by behaviour change advice to the mother to increase compliance with the intervention) on:
1. Linear growth and stunting at 18 months.
2. Cognition, nutritional status and relevant epigenetic signatures.

## METHODS AND ANALYSIS
### Study design and setting
The ENRICH study is a two-arm individually randomised open-labelled controlled trial nested within the AASH cohort study (figure 1).[25] The study will be conducted in catchment areas of three primary health centres (PHCs) (Addagutta, Warasiguda and Sitaphalmandi) in Hyderabad, India which are within a vicinity of 5 km and have similar demographic and socioeconomic composition. These slums have a total population of ≈175 000 and previous studies in this area have found a high prevalence of childhood stunting (around 27% at 24 months age).[9] It was also reported that the caloric intakes of children aged 18 months in this area were comparable to the National data of children aged 1–3 years with rural background. Women will be recruited in the study in the third trimester of pregnancy, and the egg intervention will start when the children are 9 months. Data and sample collection will be conducted as per the observational study protocol, including infant blood samples at 6 and 18 months; anthropometric measurements, dietary data and infant cognitive function at 9, 12 and 18 months of age. Blinding the participants is not possible due to

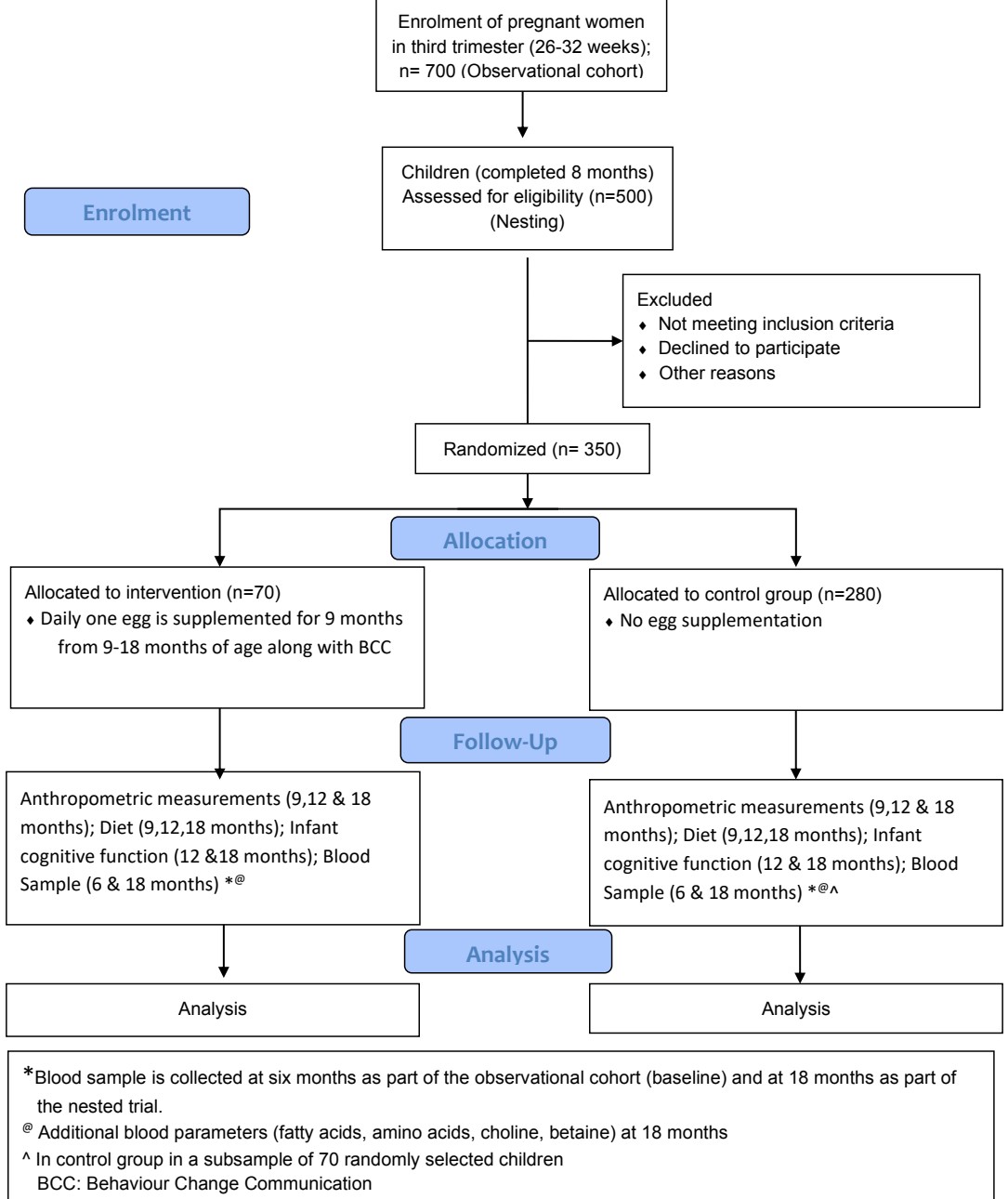

**Figure 1** CONSORT Flowchart of the study design

the nature of the intervention, but the study staff measuring study outcomes will be blinded to the intervention group.[25]

### Sample size

The sample size (70 in intervention and 280 in control group) was calculated with 90% power for an effect size of 0.32 in the length-for-age z score (LAZ) between the intervention and control groups with SD=0.70, drop-out of 10% and an allocation ratio of 1:4. This sample size is also adequate to detect 0.26 SD difference in LAZ between the two groups with 80% power and is also powered (80%) to detect a difference of 0.5 SD in the cognitive outcomes.

We believe that our study may result in a larger effect size ($\geq 0.26$ SD) than previous studies due to the intensively supervised nature of our intervention delivery which is likely to result in higher compliance to the supplement as compared with the previous studies. Our study intervention includes daily supply of ready to eat boiled egg in contrast to weekly delivery of raw eggs in Ecuador and Malawi studies,[11] [12] longer duration of

intervention (9 months vs 6 months' supply of eggs in previous studies), compliance targeted BCC including daily egg consumption record and an egg calendar. Recent studies on comprehensive supervised interventions have shown larger benefit (around 0.4 SD in LAZ) in improving linear growth faltering.[26] Moreover, an intensively supervised intervention needs to have a larger effect size to be considered for potential scale-up.

### Outcome measures

The primary study outcome is the LAZ score at 18 months. The secondary outcomes include cognition measured by the Oxford Neurodevelopment Assessment (OX-NDA) at 12 months and the INTER-NDA (a multidimensional international assessment of child development) component of the INTERGROWTH- 21st NDA. These tests have been selected since these assessments are rapid (15 min), low cost and constructed specifically for use in low-income and middle-income country setting and can be administered by non-specialists in field sites for a large sample.[27] It has strong agreement with the Bayley's Scale for Infant Development-III (interclass correlation coefficients 0.75 to 0.88, p<0.001 for all domains.[28] The OX-NDA showed moderate agreement for cognition and motor outcomes (ICCs 0.63 and 0.68, p<0.001). The tests are designed to be free from cultural biases and the kit consists of common household items. The Home Observation for Measurement of the Environment (HOME) has been widely used to assess opportunities for cognitive stimulation at home setting.[29] The scale consists of 45 items categorised into six subscales: receptivity, acceptance, organisation, learning materials, engagement and diversity. The researcher rates each item with a score of 1 if it occurred during the visit. Some items are also reported by the caregiver. Higher scores are indicative of a better environment. The scale has been previously used in India.[30 31] The other secondary outcome measurements include plasma concentrations of insulin-like growth factor 1 (IGF-1), insulin-like growth factor binding protein-3 (IGFBP3), serum/plasma concentrations of essential amino acids, micronutrient status including; retinol, folate, vitamin $B_{12}$, betaine, as well as red blood cell (RBC) fatty acids, DHA, choline and epigenetic (DNA methylation) signatures linked to the study outcomes. The detailed methodology has been described elsewhere.[25]

### Participant recruitment

The present trial is nested within the AASH observational cohort study where pregnant women (n=700) in the third trimester of pregnancy residing in the urban slums are being enrolled through the PHC after eligibility assessment and obtaining written informed consent. The data collection in pregnancy and during the follow-up of the offspring has been described elsewhere.[25] Of these, 350 children born from February 2022 to March 2023 will be a part of the present trial; 70 children will be randomised to the intervention arm and 280 to the control arm. The

control arm children will also be a part of the observational cohort.

### Eligibility

The inclusion criteria during pregnancy will be: healthy pregnant women in the age group 18–40 years, gestation 26–32 weeks, singleton pregnancy and willingness to comply with the study requirements. Children who were part of the AASH observational cohort and who attained 9 months at the inception of the ENRICH trial in November 2022 will be included. Out of these children, those who will relocate to a new area within next 9 months, those with a history of egg allergies (including family history), children with congenital anomalies or chronic morbidities, children whose parents were not willing to feed egg to their babies will be excluded from the study.

### Randomisation, intervention allocation and blinding

Each month, all children between 7.5 and 8.5 months of age will be randomly assigned to either the intervention or control group in a ratio of 1:4 after ascertaining the eligibility and obtaining written informed consent from the mother. Project staff collecting data on study outcomes will be blinded to the allocation sequence.[11]

Through a centralised block randomisation system, 350 children will be randomly allocated to blocks of 5 or 10 children, maintaining a ratio of 1:4 intervention and control children within each block. Allocation is concealed by sequentially numbered, opaque sealed envelopes.

### Trial intervention, intervention delivery and quality control

Egg supplementation will start on completion of 9 months of child age. The participant duration in the trial is for 9 months (till 18 months of age of the child). On week-days, one hen's egg (medium-sized, weighing around 50 g, raw/boiled depending on the mother's choice) will be delivered to children's homes every day by project staff, and mothers will be asked to feed it to the child within 4 hours. On fifth day, extra raw eggs are supplied to cover for the weekend days. The mothers will be provided instructions on age-appropriate egg-based complementary food recipes through a purpose-made video during the first visit. The importance of feeding the egg to the index child and not sharing it with other family members will also be stressed during monthly counselling sessions to improve adherence to the intervention. This is a model of BCC developed based on evidence generated through best practices for Take-Home Rations of supplementary nutrition by Government of India.[32] The specific messages were developed through formative research conducted with parents and grandmothers of young children. The project staff will observe the first feeding episode of complementary food containing eggs, and mothers will be counselled in case of minor gastric problems such as cramps in the stomach, nausea and vomiting.

Study staff will visit the households twice a week to observe the child feeding in the first month of the intervention and will encourage and support the mothers. Information on consumption of an egg by the child on the previous day is collected daily by the field staff while delivering the eggs in the intervention group. For comparison, data on eggs consumption will also be collected every month during a home visit in the control group children (online supplemental appendix 1).

The eggs will be procured locally from a verified supplier. The quality control measures to ensure food safety of the intervention will include periodic assessment of the microbial contamination of eggs: both at the point of purchase and the point of use. Every month, random subsamples of eggs collected soon after purchase (three raw and three boiled eggs) and complementary food prepared using eggs for the child (from six households every month) will be analysed for microbial contamination (Salmonella, Shigella, total coliform count) at the NIN laboratory. Eggs will be randomly selected for assessing nutrient composition for major nutrients (energy, proteins, iron, folate, calcium, vitamin A, D, etc) every 3 months at the food chemistry laboratory of ICMR-NIN (Indian Council of Medical Research-National Institute of Nutrition).

### Control group

The state government currently provides eggs (16 eggs per month for 7 months to 3 years) for beneficiaries of Integrated child development services (a government initiative for all women and children who can voluntarily enrol) as a part of the Rations. However, during formative research with mothers(n=100), it was found that children below 1 year of age are not fed whole eggs due to fear of indigestion and other reasons. Moreover, 59% of children were not fed eggs supplied through the government's supplementary nutrition programmes due to intrahousehold sharing.[33] Though there will not be any active intervention by the study team in the control group, all children from both the intervention and control groups may receive eggs as a part of Take-Home Ration, which may reduce the magnitude of the observed effect of the intervention. Also, there is a possibility of e 'contamination' of BCC messages in the control group in this individually randomised trial.

The number of eggs received through government programme and the number of eggs consumed by the child will be recorded for the intervention as well as control group. We will assess the mother's knowledge of the nutritional benefits of eggs in both the groups at baseline and 18 months for comparison between the two groups (online supplemental appendix 1).

### Data collection at baseline and during follow-up

Data collection in both the intervention and control groups would be done following the same protocol as the observational cohort. The parameters to be measured at different time points are listed in table 1. Details on the methodology for assessing primary and secondary outcomes are described elsewhere.[25] A trained phlebotomist will draw a 3 mL venous blood sample at both 6 months and 18 months in children from the intervention group and a subsample of 70 children from the control groups. The methods for assessment of blood parameters and epigenetic signatures linked with stunting and cognition have been described elsewhere.[22 25] Data on morbidity (including information on episodes of hospital admission and sick-child clinic visits) will be collected using a structured questionnaire every 2 weeks. Mothers will be encouraged to report any adverse events to the study staff. Data will be collected using the android interface of CommCare V.2.53 using Samsung A8 tab. The collected data will be securely stored in the database using encrypted codes. For data quality, we have used closed end questions wherever possible, and mandatory fields for core questions. Regex functionality is enabled for anthropometry data. The double data entries are manually managed at the supervisor level.

### Withdrawal from the study

During the consent process, mothers will be informed that they may choose to withdraw their child from the study at any time. Children whose mothers wish to withdraw after the randomisation will be visited by a project staff to discuss and record the reasons for withdrawal.

In case of suggestive of egg allergy symptoms, the participants would be referred to the PHC, and the mother would be advised to consult the medical officer and seek advice on the continuation of egg supplementation. Data on the prevalence of egg allergy in Indian children are not available but based on the self-reported prevalence of egg allergy (<2.5%) in other settings,[34] around 1–2 cases are anticipated.

If a participant withdraws from the trial, data captured will contribute to the analysis up to the day prior to discontinuation, unless the mother wishes to withdraw consent for any data captured up to that point. Participants who withdraw from the study will not be replaced.

### Statistical analysis

We will summarise demographic, clinical and laboratory data using descriptive statistics. For missing data, multiple imputation method will be used.[35] The primary analysis will be based on the intention-to-treat (ITT) population and will compare the children's LAZ scores at the age of 18 months in the intervention and control groups using multilevel regression models. We will adjust the models for key covariates (eg, age, gender and area) associated with the outcome variables. We will check the model assumptions and make appropriate adjustments to the analysis where necessary. We will also use difference-in-difference analysis to evaluate the impact of the intervention. We will compare the secondary outcome measures between the two groups using ITT principles. The study is powered for detecting meaningful differences in linear growth and DHA concentrations in the two groups. Other

**Table 1** Variables measured at different time points

| Measure | Timelines (intervention and control group) |
| --- | --- |
| **Biological sample/biomarkers** | |
| Buccal sample for epigenetics | At birth and at 18 months |
| Blood biomarkers: Haemoglobin, serum ferritin, serum transferrin receptors, growth hormone, serum iron, zinc, retinol, folate, vitamin $B_{12}$, folate, IGF1, IGFBP3 | At third trimester of pregnancy and at 6 and 18 months |
| Blood biomarkers relevant for egg intervention: RBC fatty acids, amino acids, choline, betaine | At 6 and 18 months (in the intervention group and in a subsample of 70 children from control group) |
| Inflammatory markers in blood: CRP and AGP | At 6 and 18 months |
| Stool sample: for entero-pathogens, parasites and inflammatory markers MPO, AAT | At 6 and 18 months |
| Micronutrient content of the breastmilk samples: vitamin A, $B_{12}$, folate | At 3 months post partum |
| **Diet, anthropometry and cognition** | |
| Diet<br>▶ Multiple pass 24-hour diet recall with repeat in 10% of the sample<br>▶ Initiation of breast feeding<br>▶ Exclusive breast feeding and introduction to complementary foods | Dietary 24-hour recall at 6, 9, 12 and 18 months<br>Breast feeding questionnaire at birth<br>Breastfeeding questionnaire at 1–6 months |
| Anthropometry measurements<br>Standing and sitting height, weight, skinfolds and MUAC<br>Length, weight, head circumference, MUAC, knee-heel length and skinfolds | At third trimester of pregnancy and at 3, 6, 9, 12, 18 and 24 months at birth. Only weight only at 1, 2, 4 and 5 months. |
| Assessment of cognition with context-specific tests<br>Home Observation Measurement of the Environment<br>Oxford Neurodevelopment Assessment (OX-NDA)<br>INTERGROWTH-21st Neurodevelopment assessment (INTER-NDA) | At 10–14 months and 18 months<br>At 10–14 months<br>At 22–24 months |
| Assessment of morbidity using structured pictorial charts | Completed by mother every day from birth until 18 months or every 2 weeks recall by the enumerator throughout the follow-up |
| Assessment of home environment | Using a structured questionnaire at 9 months |

AAT, alpha-1-antitrypsin; AGP, alpha-1-acid gycoprotein; CRP, C reactive protein; IGF1, insulin like growth factor-1; IGFBP3, insulin-like growth factor binding protein 3; MPO, myeloperoxidase; MUAC, mid upper arm circumference.

secondary outcome analyses will be exploratory. Additional analysis of the primary and secondary endpoints will also be presented using the per protocol population. No interim analysis is planned.[36 37]

### Patient and public involvement

Patients were not directly involved in the design or conduct of the study. Formative research involving mothers, grandmothers and others from the local community (excluding study participants) assisted in understanding local cultural practices and developing BCC material.[33]

The trial is registered with the Clinical Trials Registry of India (CTRI/2021/11/038208). The DSMB is independent from the sponsor and competing interests and will review clinical data at regular intervals as the study progresses.

The study results will be disseminated to the local and national government departments (for potential modifications in the supplementary nutrition programme); academic channels like national conferences and published in peer-reviewed scientific journals.

### Significance and impact

The objectives of the present trial are well aligned with the objectives of the National Nutrition Mission in India, with the potential for immediate scale-up of the intervention. The study has several strengths. Chief among them is the comprehensive evaluation of outcome indicators ranging from linear growth, cognition, nutritional biomarkers and epigenetic states indicative of exposures and biological function. Additional information on maternal diets and blood biomarkers in pregnancy as well as information on the intrauterine growth of the offspring would allow us to examine pathways for the impact that previous studies have not reported. A limitation is that in this community-based intervention, egg consumption by the index child cannot be supervised completely. Moreover, 'contamination' of the BCC messages in the control group and contamination related to the Government's supplementary nutrition programme may attenuate the effect size. Nevertheless, if the trial demonstrates significant benefits, the evidence on the impact of this relatively inexpensive yet nutrient rich ASF in early life will inform strategies for stunting reduction in India and other low/middle-income countries.

**Author affiliations**
[1]ICMR - National Institute of Nutrition, Hyderabad, India
[2]Department of Population Health, LSHTM, London, UK
[3]Epidemiology & Population Health, London School of Hygiene & Tropical Medicine, London, UK
[4]University of Aberdeen, Aberdeen, UK
[5]Department of Pathobiology and Population Sciences, University of London, London, UK
[6]London International Development Centre (LIDC), London, UK
[7]London School of Hygiene and Tropical Medicine (LSHTM), London, UK

**Collaborators** Not applicable.

**Contributors** SKB, SRC and BK: conceptualised, designed the study and wrote first draft. LFA, TD, DPP, MC, KS and PA: developed detailed protocols, planned data collection, critical input to manuscript. RM, HD-K, CH, SFR, RPullakhandam, RPalika, RRK, RNK, MLJ and PH: concept, design, critical inputs to manuscript.

**Funding** The study is funded by the UK Research and Innovation, Global Challenges Research Fund (EPPHZP5519).

**Competing interests** None declared.

**Patient and public involvement** Patients and/or the public were involved in the design, or conduct, or reporting, or dissemination plans of this research. Refer to the Methods section for further details.

**Patient consent for publication** Not applicable.

**Ethics approval** This protocol, informed consent documents and patient information sheets have been reviewed and approved by the Research Ethics Committees at ICMR-NIN (CR/3/IV/2022, dated 28 July 2022) and LSHTM (IRB no. 18029) as well as by an independent Data Safety Monitoring Board (DSMB).

**Provenance and peer review** Not commissioned; externally peer reviewed.

**Data availability statement** Data are available on reasonable request. The data from the trial will be available on request.

**ORCID iDs**
Santosh Kumar Banjara http://orcid.org/0000-0002-0893-9552
Sai Ram Challa http://orcid.org/0000-0003-3177-8133
Raghu Pullakhandam http://orcid.org/0000-0002-3758-667X
Ramachandrappa Naveen Kumar http://orcid.org/0000-0002-8984-5741
Dharani Pratyusha Palepu http://orcid.org/0000-0003-1692-020X
Preethi Gopinath http://orcid.org/0000-0003-0619-4881
Hilary Davies-Kershaw http://orcid.org/0000-0002-2044-2469
Modou Lamin Jobarteh http://orcid.org/0000-0002-7350-6980

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
