## [Reviewer comments · BMJ Paediatrics Open]

ARTICLE DETAILS

TITLE (PROVISIONAL)	Eggs for Improving Nutrition, cognitive development and Reducing linear growth retardation among Infants and young Children (ENRICH): protocol of an egg supplementation trial among 9-18 months old children in Hyderabad, India
AUTHORS	Banjara, Santosh Kumar Challa, Sai Ram Augustine, Little Flower Dasi, Teena Madhari, Radhika Fernandez Rao, Sylvia Palika, Ravindranadh Pullakhandam, Raghu Kalashikam, Rajender Rao Kumar, Ramachandrappa Naveen Palepu, Dharani Chilumula, Monica Selvaraj, Kiruthika Gopinath, Preethi Davies-Kershaw, Hilary Jobarteh, Modou Haggarty, Paul Heffernan, Claire Ferguson, Elaine Kulkarni, Bharati

VERSION 1 - REVIEW

REVIEWER	Dr. Peter Flom Peter Flom Consulting
REVIEW RETURNED	18-Feb-2023

GENERAL COMMENTS	I confine my remarks to statistical and methodological aspects of this paper. First, it's always good to read a protocol paper. I commend BMJ for publishing them! Some thoughts and questions: Why three different locations? Do you suspect that different conditions apply, even though they are all in the same city? Why match 1 to 4, condition to control? The usual is more like 1 to 1. (See e.g. https://www.researchgate.net/post/What-is-the-Rationale-for-12-ratio-in-Case-Control-studies#:~:text=Hi%2C,power%20will%20also%20be%20increase.) and using higher ratios will decrease power.
---

	"Linear growth" is a potentially confusing phrase, as it could mean two different things in this context: 1. Growth in height or 2. Straight line growth over time (that is, e.g. from 1 meter to 1.2 m to 1.4m or such over three equally spaced time points). It's pretty clear that you mean the first, but why not use the word "height"? If you suspect (and the literature review indicates that you do) that the effect of egg supplements will depend on the pre-existing diet of the kids, then you might want to include kids with varying diets. I don't know much about India, but I would guess that one way to do this would be to not choose kids exclusively in slum areas. p. 10 "Allocation is concealed until randomization" Unless I am missing something, this sentence is a mistake. Until randomization, there is no allocation to conceal. I am not sure why you are using generalized linear regression. That's not to say it's wrong, but I'd like to know why. But I was curious why, given that you have data at multiple time points, you didn't opt for something like a multilevel model, which would have greater power and let you answer more questions about growth (e.g. was it faster in the initial time period?) If you have the resources, it would also be good to have more time points, at least for height. Interesting study and I hope I get to review it after it's complete!
--	--

REVIEWER	Dr. Gabriela Montenegro Maya Health Alliance Wuqu' Kawoq, Researcher
REVIEW RETURNED	08-Mar-2023

GENERAL COMMENTS	Thank you to the authors for sharing this study protocol. As stated in the rationale, adding eggs in early complementary feeding period might be of benefit for children at risk of growth failure and retarded development. Overall I find this study interesting, however there are some aspects that I think are important to clarify before implementing the study. The authors have as a base two important studies on egg intake and growth: the Lulun (Ecuador) and the Mazira (Malawi). Both studies have linear growth as outcome, however both studies recognized limitations. Being the primary outcome length-for-age-z-scores, the sample size is too small to observe changes. I know you estimated power to detect a difference in effect size of 0.30, but why not less? Are the children of your study receiving other interventions (government/ or because they are part of another study) that might attenuate the effect? The methods are not clear to me. Participants are part of another study. Are they also part of the government program? If that is the case, what you are evaluating is not early introduction of eggs, but "frequency" of feeding. Are both groups feeding with eggs? Please clarify. I see biomarkers are taken at different time-points in the study. Is there a scientific implication of having those biomarkers at 6 months and to start the intervention at 9 months? For readers who are not familiar with India food pattern and context, are there other reasons such as veganism that makes mothers not
---

	feeding eggs to small children? I also understand the role of health workers and pediatricians who do not recommend egg intake. How do you address this? You included a series of references describing the methods. However, as readers it is important to have a brief description of methods, e.g. what are the outcomes when measuring BCC; In risk assessment, for egg allergy. What are minor gastric problems? Are there any urgent aid in case a child presents allergy? I find the study confusing on egg-delivery. As I understood, project staff will deliver 1 egg/day and present a video? Is this correct? And the staff will visit twice a week to observe the child being fed? I am not familiar with the Indian context, but I find this practice somehow exhausting for the families and for the staff, and families can feel pressured which might have an effect in the BCC outcome. Additionally, the control groups will not have the same time-exposure that might bias outcomes. If you are comparing intervention-control at least both groups should have similar behavioral exposure. Are control receiving something for their participation? In conclusion, your paper needs more description in the methodology.
--	---

REVIEWER	Dr. Elizabeth Yakes Jimenez University of New Mexico Health Sciences Center
REVIEW RETURNED	08-Mar-2023

GENERAL COMMENTS	Thank you for the opportunity to review this study protocol manuscript. I reviewed the manuscript against the SPIRIT 2013 statement/checklist (if this was used to prepare the protocol, consider mentioning in the text). Overall, the protocol is scientifically sound and the manuscript is well-written. I have a few suggestions to improve your manuscript, listed below: Introduction, paragraph 1: Consider citing this reference (https://academic.oup.com/advances/article/10/2/196/5364423) and using more nuanced language to describe the relationship between stunting and other outcomes: e.g., The high prevalence of stunting in children under five years of age in India (35%) remains a concern given the association between stunting and poor cognition, physical work capacity, and earning potential and increased risk of chronic diseases in adulthood... Introduction, paragraph 2: Review/cite more comprehensive and current assessments of the effectiveness of other interventions, and mention small quantity lipid-based nutrition supplements, e.g., https://www.thelancet.com/journals/lanchi/article/PIIS2352-4642(20)30274-1/fulltext, https://academic.oup.com/ajcn/article/114/Supplement_1/3S/6378016 Consider citing logistical/cost versus effectiveness challenges for some of these approaches relative to locally available ASF interventions. Methods and analysis, Eligibility: clarify the inclusion/exclusion criteria for the AASH cohort study versus the nested ENRICH RCT. When/how will test feeding occur (prior to enrollment)? Methods and analysis, Sample size: did you account for clustering by community (Addagutta, Warasiguda and Sitaphalmandi) in your
--

	sample size calculation, e.g., by including a design effect? If not, why not? Methods and analysis, Sample size: what difference are you powered to detect for cognition outcomes? Methods and analysis, Outcome measures: considering including more rationale as to why some of these outcome measures were selected. For example, why are you using the INTER-NDA component of the INTERGROWTH versus other available instruments to assess child cognition (e.g., validity in study population, appropriate for age group, aligns with expected impact, etc.)? I think it is helpful to at least briefly include this rationale in the protocol manuscript versus pointing the reader to other publications. Methods and analysis, Trial intervention, intervention delivery, and quality control: State the duration of the egg supplementation (until 18 months of age). How many eggs will be assessed for nutrient composition at each timepoint, and how will these eggs be selected? Methods and analysis, control group: I found these sentences confusing - "Therefore, our intervention will include a strong BCC focusing on the nutritional benefits of feeding whole eggs to young children. There will not be any active intervention by the study team in the control group." You mix the discussion of the rationale and delivery for the intervention and control groups across the two intervention/control paragraphs (for example, you also mention "...data on eggs consumption will also be collected every month during a home visit in the control group children" in the intervention paragraph). Since the formative research is relevant to how you designed both groups, maybe discuss separately, first? Then describe just the delivery for each group. Methods and analysis, control group: Consider discussing the potential for bias due to the large difference in study staff contact time for individuals in the intervention vs. control groups (daily versus monthly). Methods and analysis, data management: the SPIRIT checklist recommends reporting "Plans for data entry, coding, security, and storage, including any related processes to promote data quality (eg, double data entry; range checks for data values). Reference to where details of data management procedures can be found, if not in the protocol." Also consider addressing access to data. Methods and analysis, statistical analysis: will you use multilevel models to account for clustering by community? If not, why not (e.g., ICC negligible/not different from zero)? See, for example, the discussion here http://www.bristol.ac.uk/cmm/learning/multilevel-models/what-why.html and general steps to consider in this article, Figure 6 (https://rips-irsp.com/articles/10.5334/irsp.90). Methods and analysis, statistical analysis: how will you handle missing data? Table 1: Consider using additional table subheaders or bolding to make this table a little bit easier to read and follow. Not necessary to list multiple pass 24 hour recalls twice – just include infant timepoints on that line as well. Infant timepoints don't line up with infant measures under anthropometry. Need footers to define some
--	---

	acronyms (e.g., MUAC, IGF1, CRP, AGP, MPO, AAT). Will you use a specific instrument to assess home environment (e.g., Family Care Indicators)? If not, what will be assessed in the home environment?
--	---

VERSION 1 – AUTHOR RESPONSE

Reviewer: 1

Why three different locations? Do you suspect that different conditions apply, even though they are all in the same city?

Response: If a single location is targeted, the time required to recruit the mother child dyads will prolong the study period. Hence three locations which are near contiguous primary health care areas are considered They are within a vicinity of 5 km with similar demographic and socioeconomic composition.

Page no. 6, lines 165-166

Why match 1 to 4, condition to control? The usual is more like 1 to 1. (See e.g. <https://www.researchgate.net/post/What-is-the-Rationale-for-12-ratio-in-Case-Control-studies#:~:text=Hi%2C,power%20will%20also%20be%20increase.>) and using higher ratios will decrease power.

Response: We are using Blocked Randomization with Randomly Selected Block Sizes which should address this problem (Kaltenbach HM. Chapter 7 Improving Precision and Power: Blocked Designs | Statistical Design and Analysis of Biological Experiments [Internet]. [cited 2023 Apr 7]. Available from: <https://n.ethz.ch/~kahans/doe2021/ch-blocking.html>

“Linear growth” is a potentially confusing phrase, as it could mean two different things in this context: 1. Growth in height or 2. Straight line growth over time (that is, e.g. from 1 meter to 1.2 m to 1.4m or such over three equally spaced time points). It’s pretty clear that you mean the first, but why not use the word “height”?

Response: The standard for linear growth has a part based on length (length-for-age, 0 to 24 months) and another on height (height-for-age, 2 to 5 years). The two parts were constructed using the same model but the final curves reflect the average difference between recumbent length and standing height.

Group WMGRS, de Onis M. WHO Child Growth Standards based on length/height, weight and age. *Acta Paediatrica*. 2006;95(S450):76–85.

If you suspect (and the literature review indicates that you do) that the effect of egg supplements will depend on the pre-existing diet of the kids, then you might want to include kids with varying diets. I don't know much about India, but I would guess that one way to do this would be to not choose kids exclusively in slum areas.

Response: To establish the proof of concept, we believe that it will be helpful to have a group of kids with more or less similar diets which will help us explore the effect of an additional egg to the diet when the children have similar diets. However, in a previous study in the same area, it was reported that the caloric intakes of 18 month old children in this area were low (590 ± 282.8 Kcal), but were comparable to the National data of 1-3 year old children with rural background.

Ref.No. 8 Radhakrishna et al., Effectiveness of zinc supplementation to full term normal infants: a community based double blind, randomized, controlled, clinical trial. *PLoS One*. 2013 May 30;8(5):e61486. doi: 10.1371/journal.pone.0061486. PMID: 23737940; PMCID: PMC3667840.

p. 10 "Allocation is concealed until randomization" Unless I am missing something, this sentence is a mistake. Until randomization, there is no allocation to conceal.

Response: To spare the number of words the statement was written similar to standard way of reporting in protocol papers viz "Recruitment of 1240 patients will occur over 3 years with allocation concealment until randomisation by a centralised service." From the paper 10.1136/bmjopen-2016-015291

The sentence is being rephrased based on the exact method we are following as "Allocation is concealed by sequentially numbered, opaque sealed envelopes (SNOSE) method" Reference <https://doi.org/10.1016/j.jcrc.2005.04.005>

Page 8, line 240

I am not sure why you are using generalized linear regression. That's not to say it's wrong, but I'd like to know why. But I was curious why, given that you have data at multiple time points, you didn't opt for something like a multilevel model, which would have greater power and let you answer more questions about growth (e.g. was it faster in the initial time period?)

If you have the resources, it would also be good to have more time points, at least for height.

Response: These time-points have been developed in line with the funding time-line.

Interesting study and I hope I get to review it after it's complete!

Response: Thank you

Reviewer: 2

Thank you to the authors for sharing this study protocol. As stated in the rationale, adding eggs in early complementary feeding period might be of benefit for children at risk of growth failure and retarded development. Overall I find this study interesting, however there are some aspects that I think are important to clarify before implementing the study.

Response: We thank the reviewer for the positive comment.

The authors have as a base two important studies on egg intake and growth: the Lulun (Ecuador) and the Mazira (Malawi). Both studies have linear growth as outcome, however both studies recognized limitations. Being the primary outcome length-for-age-z-scores, the sample size is too small to observe changes. I know you estimated power to detect a difference in effect size of 0.30, but why not less? Are the children of your study receiving other interventions (government/ or because they are part of another study) that might attenuate the effect?

Response: We request the reviewer to kindly refer to the previously answered comment in section 2.1

The methods are not clear to me. Participants are part of another study. Are they also part of the government program? If that is the case, what you are evaluating is not early introduction of eggs, but "frequency" of feeding. Are both groups feeding with eggs? Please clarify.

Response: Participants, (both intervention and control arms) are drawn from an observational cohort study, (The Action Against Stunting Study) . The intervention group will be sequestered from the main cohort. All participants are part of the ongoing government program where 16 eggs are given as take home rations (THR). However due to lack of strong BCC the natural sequence of the program is under coverage and low compliance. As informed in the manuscript, during formative research with mothers, it was found that children below one year of age are not fed whole eggs due to fear of

indigestion and other reasons. Moreover, 59% of children were not fed eggs supplied through the government's supplementary nutrition programs due to intra-household sharing. The early intervention at 9 months along with daily whole egg consumption is being enforced through strong BCC in the intervention group. However, the background rate of consumption of the eggs supplied through government program is also being recorded. This data will be included in the data analysis.

I see biomarkers are taken at different time-points in the study. Is there a scientific implication of having those biomarkers at 6 months and to start the intervention at 9 months?

Response: The 6-month and 18-month time point for biomarker is as per the ongoing cohort study (AASH). Additional blood sample collection from children again at 9 months is challenging to the child, parent and the investigators and would require an increase in budget. Scientific implication cannot be completely ruled out and is a limitation of the study. However, the time between 6 months and 9 months is part of the weaning phase and as per our formative study eggs are rarely introduced during this time period.

For readers who are not familiar with India food pattern and context, are there other reasons such as veganism that makes mothers not feeding eggs to small children? I also understand the role of health workers and pediatricians who do not recommend egg intake. How do you address this?

Response: Vegetarianism is prevalent in India and not veganism. Please see section 4.3, where we have addressed the rest of your comment.

This is a good point and we can do a separate study to assess if health workers and paediatricians do not recommend egg intakes or are there any physician induced food fads.

You included a series of references describing the methods. However, as readers it is important to have a brief description of methods, e.g. what are the outcomes when measuring BCC;

Response: The baseline, midline and endline Knowledge of the caregivers apart from compliance rates of whole eggs constitute the outcome measures of BCC. Page 9. Lines 285-287

In risk assessment, for egg allergy. What are minor gastric problems? Are there any urgent aid in case a child presents allergy?

Response: Minor gastric problems include digestive symptoms, such as cramps, nausea and vomiting. Addition, Page 9, lines 255-256.

Since egg allergies typically present immediately or within two hours after ingestion, an auxiliary nurse midwife will be present during the first egg feed and she will observe the child for any for signs of egg allergies or anaphylaxis for a period of 2 hours. In case of allergy/anaphylaxis, allergies will be clinically evaluated by a paediatrician and treated clinically with a referral to tertiary hospital for management after giving a dose of antihistamine in the field unit of the study.

Second generation antihistamines and the dose will be kept with the Staff nurse for use in case of emergency.

-Cetirizine 6 months to < 2 years: 0.25 mg/kg;

- Levocetirizine 6 months to 5 years: 1.25 mg OD

Any child exhibiting egg allergy will be excluded from the study (Appendix 2)

I find the study confusing on egg-delivery. As I understood, project staff will deliver 1 egg/day and present a video? Is this correct? And the staff will visit twice a week to observe the child being fed? I am not familiar with the Indian context, but I find this practice somehow exhausting for the families

and for the staff, and families can feel pressured which might have an effect in the BCC outcome. Additionally, the control groups will not have the same time-exposure that might bias outcomes. If you are comparing intervention-control at least both groups should have similar behavioral exposure. Are control receiving something for their participation?

Response: Project staff will deliver 1 egg per day. Monthly visits are done by the team members to deliver the BCC component which consist of one video, 2 testimonials, a booklet explaining the importance of egg for children, one recipe guidance book to explain to the participants regarding adding variety to the egg feeding. These aids are used during monthly counselling session by the staff. The control group will be provided a small gift such as a tiffin box/bowl which is equivalent to the amount spent on eggs but do not influence the outcome.

Reviewer: 3

Thank you for the opportunity to review this study protocol manuscript. I reviewed the manuscript against the SPIRIT 2013 statement/checklist (if this was used to prepare the protocol, consider mentioning in the text). Overall, the protocol is scientifically sound and the manuscript is well-written. I have a few suggestions to improve your manuscript, listed below:

Response: The authors wish to thank the reviewer for this motivating comment.

Introduction, paragraph 1: Consider citing this reference (<https://academic.oup.com/advances/article/10/2/196/5364423>) and using more nuanced language to describe the relationship between stunting and other outcomes: e.g., The high prevalence of stunting in children under five years of age in India (35%) remains a concern given the association between stunting and poor cognition, physical work capacity, and earning potential and increased risk of chronic diseases in adulthood...

Response: We have included the following reference on page 1, line 98

Jef L Leroy, Edward A Frongillo, Perspective: What Does Stunting Really Mean? A Critical Review of the Evidence, *Advances in Nutrition*, Volume 10, Issue 2, March 2019, Pages 196–204, <https://doi.org/10.1093/advances/nmy101>

Introduction, paragraph 2: Review/cite more comprehensive and current assessments of the effectiveness of other interventions, and mention small quantity lipid-based nutrition supplements, e.g., [https://www.thelancet.com/journals/lanchi/article/PIIS2352-4642\(20\)30274-1/fulltext](https://www.thelancet.com/journals/lanchi/article/PIIS2352-4642(20)30274-1/fulltext),

Response: We have included the following reference on page 1, line 121-123

Keats EC, Das JK, Salam RA, Lassi ZS, Imdad A, Black RE, Bhutta ZA. Effective interventions to address maternal and child malnutrition: an update of the evidence. *Lancet Child Adolesc Health*. 2021 May;5(5):367-384. doi: 10.1016/S2352-4642(20)30274-1. Epub 2021 Mar 7. PMID: 33691083.

Consider citing logistical/cost versus effectiveness challenges for some of these approaches relative to locally available ASF interventions.

Response: We could not find any relevant papers to cite in this area. However, a paper citing that eggs are one of the cheapest source of animal source food has been cited as ref 15, page 4, line 122-124.

Methods and analysis, Eligibility: clarify the inclusion/exclusion criteria for the AASH cohort study versus the nested ENRICH RCT. When/how will test feeding occur (prior to enrollment)?

Response: Children who were part of the AASH observational cohort and who attained 9 months at the inception of the ENRICH trial in November 2022 were included. Out of these children, those who

would relocate to a new area within next 9 months, those with a history of egg allergies (including family history), children with congenital anomalies or chronic morbidities, children whose parents were not willing to feed egg to their babies were excluded from the study. The test feeding was done only for children who were enrolled but have not been given egg previously and was done in the presence of a field medical officer. The team observed the child for half an hour and provided contact number to the caregivers to contact the study team in case of an emergency. On trial feeding days, the team kept themselves available in the field station to reach out to the participants in case of any emergency. Added in page no.8, lines 225-230

Methods and analysis, Sample size: did you account for clustering by community (Addagutta, Warasiguda and Sitaphalmandi) in your sample size calculation, e.g., by including a design effect? If not, why not?

Response: Please refer to section 3.3 for our response to this comment

Methods and analysis, Sample size: what difference are you powered to detect for cognition outcomes?

Response: The power for detecting differences between the intervention and control groups determined by assuming that $\alpha = 0.05$, (two-tailed tests) and power of ≥ 0.80 to detect a difference of 0.50 SDs a sample size of 63 is required per group. Addition: Page 6, line 182-183

Methods and analysis, Outcome measures: considering including more rationale as to why some of these outcome measures were selected. For example, why are you using the INTER-NDA component of the INTERGROWTH versus other available instruments to assess child cognition (e.g., validity in study population, appropriate for age group, aligns with expected impact, etc.)? I think it is helpful to at least briefly include this rationale in the protocol manuscript versus pointing the reader to other publications.

Response: The Oxford Neurodevelopment Assessment (OX-NDA) and The Intergrowth-21st Neurodevelopment Assessment (INTER-NDA) are neurodevelopment assessments of one (11-13 months), and two-year-olds (21-30 months) respectively. The test has detailed validation using multiple statistical techniques, in a population-based sample from LMIC setting. The OX-NDA has strong positive correlations (Pearson's $r=0.50-0.52$, $p<0.001$) with cognition and motor scores of the Bayley Scales of Infant Development-III a gold standard for assessment of infant development and the INTER-NDA has demonstrated strong agreement with the BSID-III (interclass correlation coefficients 0.75 to 0.88, $p<0.001$ for all domains with little to no bias on Bland Altman analysis). The tests are designed to be free from cultural biases and the kit consists of common household items. The INTER-NDA has been translated into few languages including, the Marathi language of India the translation included processes of cultural customization, translation and back translation. The assessments are rapid (15 minutes), low-cost, and constructed specifically for use in low and middle-income (LMIC) setting. Both tests have 37 items grouped into cognition, motor, language, and positive and negative behaviour domains and can be administered by non-specialists. Page 7, Lines 198-204

Methods and analysis, Trial intervention, intervention delivery, and quality control: State the duration of the egg supplementation (until 18 months of age). How many eggs will be assessed for nutrient composition at each timepoint, and how will these eggs be selected?

Response: The participant duration for the ENRICH study is 9 months, The trial duration is till May 2024. Eggs will be randomly selected for assessing nutrient composition for major nutrients (energy, proteins, iron, folate, calcium, vitamin A, D etc) every three months at the food chemistry laboratory of ICMR- NIN.

Page 9, lines 266-267

Methods and analysis, control group: I found these sentences confusing - "Therefore, our intervention will include a strong BCC focusing on the nutritional benefits of feeding whole eggs to young children. There will not be any active intervention by the study team in the control group." You mix the discussion of the rationale and delivery for the intervention and control groups across the two intervention/control paragraphs (for example, you also mention "...data on eggs consumption will also be collected every month during a home visit in the control group children" in the intervention paragraph). Since the formative research is relevant to how you designed both groups, maybe discuss separately, first? Then describe just the delivery for each group.

Response: The paragraph has been restructured to reflect the suggestions given by the reviewer Line 270-284, Page 9.

Methods and analysis, control group: Consider discussing the potential for bias due to the large difference in study staff contact time for individuals in the intervention vs. control groups (daily versus monthly).

Response: The authors agree to the fact that there is a large difference in study staff contact time for individuals in intervention and control group. However considering that communication/ behaviour communication is also part of the intervention, a performance bias may not be attributed here.

Ref. Mansournia MA, Higgins JP, Sterne JA, Hernán MA. Biases in Randomized Trials: A Conversation Between Trialists and Epidemiologists. *Epidemiology*. 2017 Jan;28(1):54-59. doi: 10.1097/EDE.0000000000000564. Erratum in: *Epidemiology*. 2018 Sep;29(5):e49. PMID: 27748683; PMCID: PMC5130591

Methods and analysis, data management: the SPIRIT checklist recommends reporting "Plans for data entry, coding, security, and storage, including any related processes to promote data quality (eg, double data entry; range checks for data values). Reference to where details of data management procedures can be found, if not in the protocol." Also consider addressing access to data.

Response: Data will be collected using the android interface of commcare version 2.53 using Samsung A8 tab. The collected data is securely stored in the database using encrypted codes which is located in the India. For data quality, we have used closed end questions wherever possible, and mandatory fields for core questions wherein the enumerator cannot move forward without responding to the particular question, In addition, regex functionality is enabled for anthropometry data. The double data entries are manually managed at the supervisor level. Page 10, lines 301-305

Methods and analysis, statistical analysis: will you use multilevel models to account for clustering by community? If not, why not (e.g., ICC negligible/not different from zero)? See, for example, the discussion here <http://www.bristol.ac.uk/cmm/learning/multilevel-models/what-why.html> and general steps to consider in this article, Figure 6 (<https://rips-irsp.com/articles/10.5334/irsp.90>).

Response: No, all the three communities (areas) are in the same city with almost similar geographical / demographical characteristics. We are selecting 3 areas just to get expected sample size. We are not expecting any difference in the outcomes between the communities. If we get any difference then we will plan for multilevel models for clustering by community. This has been inserted in page 6, 164-165.

Methods and analysis, statistical analysis: how will you handle missing data?

Response: We intend to prevent the problem by planning the study and collecting the data carefully. We train all personnel related to the study on all aspects of the study, such as the participant enrolment, collection and entry of data, and implementation of the intervention. The data collection in the CAPI is monitored real-time by the data manager through dashboard and efforts made to

minimise missing of data. In spite of all these efforts if the data is missed, we will try to administer the Multiple imputation method described by Kang 201

Page 11, line 324

(Kang H. The prevention and handling of the missing data. Korean J Anesthesiol. 2013 May;64(5):402–6.)

Table 1: Consider using additional table subheaders or bolding to make this table a little bit easier to read and follow. Not necessary to list multiple pass 24 hour recalls twice – just include infant timepoints on that line as well. Infant timepoints don't line up with infant measures under anthropometry. Need footers to define some acronyms (e.g., MUAC, IGF1, CRP, AGP, MPO, AAT). Will you use a specific instrument to assess home environment (e.g., Family Care Indicators)? If not, what will be assessed in the home environment?

Response: Modified Table 1.

VERSION 2 – REVIEW

REVIEWER	Dr. Peter Flom Peter Flom Consulting
REVIEW RETURNED	18-May-2023

GENERAL COMMENTS	The authors have addressed my concerns and I now recommend publication
--

REVIEWER	Dr. Gabriela Montenegro Maya Health Alliance Wuqu' Kawoq, Researcher
REVIEW RETURNED	22-May-2023

GENERAL COMMENTS	Thank you to the Authors for submitting the revised version of the MSC. You have addressed the different comments for the reviewers and it has helped improving the paper. There are still some minor comments that need to be clarified. I am looking forward to see published results. Line 198- take out the parenthesis – “(The INTER-NDA...” I find inconsistencies in the assessment in cognition. Please clarify. Please add one or two references to the validity of the INTER-NDA scale. I noticed that in Table 1 you described using other assessments at different time points: a) Home Observation Measurement of the Environment (HOME - Scale) b) Oxford Neurodevelopment Assessment (OX-NDA) c) And INTER-NDA. Are all the same instrument? Otherwise please add them to the Methods section. Additionally the study ends at 18 months? And the Table 1 indicates that scales are used at different age. If it is correct please do this clarification in the Methods section. Table 1- Morbidity section. It says daily and then every two weeks. Figure 1- add diet to the follow-up visits. Line 243- I still find confusing the eggs delivery, in the intervention they receive daily the visit? Including weekends? So they will have an average home visit of 30 days and prox. 30 eggs/ month. Is it correct? The intervention is 9 months. Every month they will have
---

	counseling, is this correct? If so please describe it so it is simple to understand.
--	--

REVIEWER	Dr. Elizabeth Yakes Jimenez University of New Mexico Health Sciences Center
REVIEW RETURNED	31-May-2023

GENERAL COMMENTS	Thank you for the opportunity to review the revision of this manuscript and for the effort that the authors put towards responding to each comment. I believe that the revisions are adequate and recommend acceptance for publication.
---

VERSION 2 – AUTHOR RESPONSE

Reviewer: 1

The authors have addressed my concerns and I now recommend publication

Response: Thank you.

Reviewer: 2

Thank you to the Authors for submitting the revised version of the MSC. You have addressed the different comments for the reviewers and it has helped improving the paper. There are still some minor comments that need to be clarified. I am looking forward to see published results.

Line 198- take out the parenthesis - "(The INTER-NDA...":

Response: Done.

I find inconsistencies in the assessment in cognition. Please clarify.

Response: The rationale for choosing the measures was previously reported in a systematic literature review conducted at the beginning of the project (Munoz-Chereau, Ang, Dockrell, Outhwaite, & Heffernan, 2021). Through that review suitable measures based on 1. psychometric quality (validity and reliability), 2. cultural adaptability and 3. the use in the field sites for a large sample were identified. Reference and text added in page 7, line 201-204.

Please add one or two references to the validity of the INTER-NDA scale.

Response: Reference added: Fernandes M, Villar J, Stein A, et al I (2020). INTERGROWTH-21st Project international INTER-NDA standards for child development at 2 years of age: an international prospective population-based study BMJ Open ;10:e035258. doi: 10.1136/bmjopen-2019-035258

I noticed that in Table 1 you described using other assessments at different time points:

- a) Home Observation Measurement of the Environment (HOME -Scale)
- b) Oxford Neurodevelopment Assessment (OX-NDA)
- c) And INTER-NDA. Are all the same instrument? Otherwise please add them to the Methods section. Additionally the study ends at 18 months? And the Table 1 indicates that scales are used at different age. If it is correct please do this clarification in the Methods section.

Response: The INTER-NDA will be used for measuring development at 22 months. The INTER -NDA has been validated for use in children ages 22 to 30 months, however it includes items for children ages 18 to 36 months. The same scale will be used for the main cohort of infants. The INTERNDA was validated in India, Kenya, Italy, Brazil, and the UK (Fernandes et al (2020)). The OX-NDA is being used for measuring development at 12 months among the participant children of the observational cohort. Because the study is a nested trial, this measurement will be available for the ENRICH participants as well.

Corrections are done in the table and is indicated in red font. Details regarding OX-NDA and HOME has been added to the manuscript method section, page 7, lines 198 -214.

Table 1- Morbidity section. It says daily and then every two weeks.

Response: The pictorial chart consist of option to record morbidity daily by the caregiver. After two weeks this data is collected by the enumerator. If the pictorial chart is not completed by the caregiver, the enumerator fills the data based on a recall by the care-giver. The same has been indicated in table 1.

Figure 1- add diet to the follow-up visits.

Response: Done

Line 243- I still find confusing the eggs delivery, in the intervention they receive daily the visit? Including weekends? So they will have an average home visit of 30 days and prox. 30 eggs/ month. Is it correct? The intervention is 9 months. Every month they will have counseling, is this correct? If so please describe it so it is simple to understand.

Response : Yes. The intervention group will receive a daily visit on weekdays. On 5th weekday, they will be additionally provided the eggs which need to be fed to the child on Saturday and Sunday. These additional eggs will be raw eggs. The average home visit would be 22 days and the eggs received would be 30 eggs/month?. Every month the care-givers of the intervention group receive counseling on importance of feeding egg to the child. Modifications to indicate these corrections are done in page 9, lines 254-258.

Reviewer: 3

Thank you for the opportunity to review the revision of this manuscript and for the effort that the authors put towards responding to each comment. I believe that the revisions are adequate and recommend acceptance for publication.

Response: Thank you!